# Identification of Key Prognostic Genes of Triple Negative Breast Cancer by LASSO-Based Machine Learning and Bioinformatics Analysis

**DOI:** 10.3390/genes13050902

**Published:** 2022-05-18

**Authors:** De-Lun Chen, Jia-Hua Cai, Charles C. N. Wang

**Affiliations:** 1Department of Bioinformatics and Medical Engineering, Asia University, Taichung 41354, Taiwan; delunchen.0509@gmail.com; 2Institute of Statistical Science, Academia Sinica, Taipei 11529, Taiwan; as6309123@yahoo.com.tw; 3Center for Precision Health Research, Asia University, Taichung 41354, Taiwan

**Keywords:** triple negative breast cancer, differentially co-expressed genes, bioinformatics analysis, biomarkers

## Abstract

Improved insight into the molecular mechanisms of triple negative breast cancer (TNBC) is required to predict prognosis and develop a new therapeutic strategy for targeted genes. The aim of this study is to identify key genes which may affect the prognosis of TNBC patients by bioinformatic analysis. In our study, the RNA sequencing (RNA-seq) expression data of 116 breast cancer lacking ER, PR, and HER2 expression and 113 normal tissues were downloaded from The Cancer Genome Atlas (TCGA). We screened out 147 differentially co-expressed genes in TNBC compared to non-cancerous tissue samples by using weighted gene co-expression network analysis (WGCNA) and differential gene expression analysis. Then, Gene Ontology (GO) and Kyoto Encyclopedia of Genes and Genomes (KEGG) pathway enrichment analyses were constructed, revealing that 147 genes were mainly enriched in nuclear division, chromosomal region, ATPase activity, and cell cycle signaling. After using Cytoscape software for protein-protein interaction (PPI) network analysis and LASSO feature selection, a total of fifteen key genes were identified. Among them, BUB1 and CENPF were significantly correlated with the overall survival rate (OS) difference of TNBC patients (*p* value < 0.05). In addition, BUB1, CCNA2, and PACC1 showed significant poor disease-free survival (DFS) in TNBC patients (*p* value < 0.05), and may serve as candidate biomarkers in TNBC diagnosis. Thus, our results collectively suggest that BUB1, CCNA2, and PACC1 genes could play important roles in the progression of TNBC and provide attractive therapeutic targets.

## 1. Introduction

Triple negative breast cancer (TNBC) is a subtype of breast cancer that lacks expression of the estrogen receptor (ER), progesterone receptor (PR), and human epidermal growth factor receptor 2 (HER2) following immunohistochemical analysis [1]. It is estimated that approximately 200,000 cases are diagnosed with TNBC among women worldwide, accounting for 15% of all breast cancers each year [2]. Although there are many types of cancer treatments, including surgeries, chemotherapies, and radiation therapies, approved for patients with TNBC, it remains a significantly poorer prognosis with lower survival rates than other types of breast cancer. Early detection and treatments are crucial for improving cancer outcomes [3,4].

The rapid development of microarray and next-generation sequencing technologies provides researchers with the ability to detect changes in gene expression data of various cancer types [5]. Weighted gene co-expression network analysis (WGCNA), also known as weighted correlation network analysis, is a systematic biology approach to describe the correlation of gene expression between different samples. This approach has been widely used to find highly relevant gene clusters (modules) and identify candidate hub genes based on the interconnectivity of gene modules and the association between gene modules and clinical traits [6]. LASSO is one of the common machine learning methods. It can effectively select important feature values with non-zero coefficients through regularization. Therefore, it is widely used in the classification or feature selection of high-dimensional data [7,8].

In our study, WGCNA integrated with differential gene expression analysis was applied to analyze high-throughput sequencing data from the TCGA database to identify differentially co-expressed genes between TNBC and normal tissues samples, followed by Gene Ontology (GO) and Kyoto Encyclopedia of Genes and Genomes (KEGG) pathway enrichment analyses. Moreover, a protein–protein interaction network was established by STRING, and candidate key genes were identified by the CytoHubba plug-in in Cytoscape and a LASSO regression model. Finally, we verified the prognostic value of key genes based on data from Kaplan–Meier (KM) Plotter databases [9]. These findings may be important in evaluating the malignant potential and clinical outcomes of patients with TNBC.

## 2. Materials and Methods

### 2.1. Data Collection from TCGA

The workflow of this study is shown in Figure 1. The whole transcriptome sequencing data set of raw read counts and fragments per kilobase per million (FPKM) values, and corresponding clinical data (Appendix A) of all breast cancer patients were downloaded from TCGA-BRCA dataset using R package *TCGAbiolinks* (V2.14.1) [10]. The raw read counts are used for the analysis of differentially expressed genes, while FPKM data were used for WGCNA and subsequent downstream analyses. We collected 1222 breast cancer samples, including 1109 tumor and 113 normal tissue samples, from the TCGA database. Based on the clinical information of the samples, 116 TNBC samples with the lack of expression in ER, PR, and HER2 and 113 normal breast tissue samples were retained in this study for further analysis. The demographic information of the 229 samples is shown in Appendix A.

### 2.2. Screening for Differentially Expressed Genes

Differential gene expression analysis provides a method to identify the changes of gene expression between experimental and control groups. In the present study, the R package *DESeq2* (V1.26.0) [11] was applied to screen the differentially expressed genes (DEGs) in selected TNBC compared with normal samples. The gene names corresponding to the Ensemble ids were converted into a gene symbol via the Ensembl database. For the repetitive gene symbols, only the one with the highest sum of expression levels is kept. Before analyzing DEGs, *DESeq2* will use the median of ratios method to perform normalization preprocessing on the input raw read counts data. Subsequently, genes whose total expression level is less than 1 in all samples are deleted. The *p*-values of DEGs were corrected to false discovery rate (FDR) following the Benjamini–Hochberg method [12]. DEGs with FDR < 0.05 and | log2 fold change (FC)| > 1 were considered a statistically significant difference. Finally, 9226 DEGs were visualized as a volcano plot using R package *ggplot2* (V3.3.0) [13], and used further as candidate genes for co-expression network construction.

### 2.3. Construction of Co-Expression Network and Identificaion of Key Modules

WGCNA is used to explore gene modules highly related to the external traits of the samples. The weighted gene co-expression network of 9226 DEGs in our study was constructed through the R package *WGCNA* (V1.69) [6]. The correlation coefficients between genes were calculated using the following formula: $aij={\left|Sij\right|}^{\beta }$ (aij: Adjacency between gene *i* and gene *j*, Sij: Pearson’s correlation between gene i and gene j, β: Soft threshold). Then, the adjacency matrix was converted into a topological overlap matrix (TOM). In WGCNA, the linkage hierarchical clustering was carried out for the genes dendrogram based on dissimilarity measure (1-TOM), and the minimum size (gene group) was set as 150 in order to classify the genes with similar expression profiles into the same gene module.

To further determine the key modules in the co-expression network, the relevance of module eigengene (ME) to clinical traits was calculated, and the association of each gene with clinical significance was measured by gene significance (GS). Module significance (MS) is considered as the average GS for all genes in a module. The modules that typically have the highest absolute MS ranking among all modules were selected for further analysis.

### 2.4. Candidate Module Hub Genes Identification

After identifying significant interest of modules, GS and module membership (MM) were calculated for each gene. In WGCNA, MM refers to the correlation between genes and gene expression profiles. Hub genes are a subset of highly interconnected genes (nodes) within key modules of co-expression network, and are significantly associated with biological functions. In our study, to further identify genes related to TNBC, we selected genes with |GS| > 0.2 and |MM| > 0.8 as module hub genes.

### 2.5. Functional and Pathway Enrichment Analyses

To understand the function of hub genes in the key modules, GO and KEGG pathways were implemented using R package *clusterProfiler* (V3.14.3) [14]. The *clusterProfiler* uses Bioconductor GO.db (V3.14) and KEGG.db (V3.2.4) to annotate genes with GO and KEGG terms. In GO analysis, GO terms contain biological process (BP), cellular component (CC), and molecular function (MF) [15]. In addition to the function annotation of the genes, KEGG is helpful to clarify the signaling transduction pathway involved in hub genes [16]. A *p*-value < 0.05 was set as the cut-off standard.

### 2.6. PPI Network Construction and Modules Selection

The online STRING database (https://string-db.org/, V11.0) (accessed on 18 December 2021) is used to predict and analyze the functional association network between proteins in the organism [17]. In this study, the STRING database was applied for constructing the PPI network of hub genes, and the PPI network with functional association score ≥ 0.7 was visualized based on Cytoscape (https://cytoscape.org/, V3.8.0) (accessed on 18 December 2021) [18].

Subsequently, the hub genes of the PPI network were screened by using the CytoHubba (V0.1) plug-in in Cytoscape according to the maximum clique centrality (MCC) score of each node. The MCC algorithm formula is as follows: $MCC\left(v\right)=\sum_{C\in S\left(v\right)}\left(\left|\;C\;\right|-1\right)!$ (S(v): The collection of Maximal Clique containing node v, (| C |−1)!: is the product of all positive integers less than | C |). The 10 genes with the highest MCC score are selected as the hub genes. As previously reported, MCC algorithm can effectively find the important nodes in the network, and the nodes with the top 10 MCC scores have been proven to be effective [19].

### 2.7. Feature Selection by LASSO Regression

LASSO regression analysis was performed using the R package glmnet (V4.1) [20], and LASSO regression analysis methods are often used for feature selection or as a binary classifier. In this study, a combination of candidate genes associated with TNBC was selected using the features of LASSO feature selection. Cross-validation can effectively improve the performance of the model, so we use 10-fold cross-validation to train the model. Lambda 1 standard error (lambda.1se) usually optimizes regularization, which keeps the error and the minimum error within one standard deviation error [21].

### 2.8. Verification of Prognostic Value of Hub Genes

To verify the prognostic value of hub genes and significant genes, OS and DFS prognostic analyses were performed based on the expression and clinical data from KM Plotter (http://www.kmplot.com) (accessed on 18 December 2021) databases. KM Plotter used Cox proportional hazards regression were used to perform prognostic analysis for each gene, and the Benjamini–Hochberg method was used to correct for multiple hypothesis testing [9,22]. We divided the patients into high and low expression groups according to their average expression value of hub genes, and the *p*-value of the prognostic analysis results was corrected by FDR. The log-rank *p*-values < 0.05 were considered to be statistically significant.

## 3. Results

### 3.1. Identification of DEGs in TNBC from TCGA

The DEGs analysis was performed on the RNA-seq data of 116 TNBC and 113 normal tissue samples from TCGA database. In this study, we screened out a total of 9226 DEGs, including 5626 up-regulated and 3600 down-regulated genes, based on the screening criteria of FDR < 0.05 and |log2 FC| > 1. The volcano plot of 9226 DEGs is shown in Figure 2.

### 3.2. Weighted Co-Expression Network Construction and Gene Modules Identification

We utilized the *WGCNA* package to construct a co-expression network for 9226 DEGs, and those with similar expression patterns were grouped into the same gene module through average linkage clustering. In choosing the best thresholds, the network topology for soft-thresholding powers from 1 to 20 was calculated by scale independence and mean connectivity analysis of modules (Figure 3a,b). As shown in Figure 3c,d, when the soft threshold β was selected as 5, the power for the scale-free topology fit index reached 0.95.

To ensure compliance with low correlation, the dissimilarity between each module was evaluated. The modules with the dissimilarity < 0.2 were subsequently merged (Figure 4a), resulting in a total of 14 modules in this study (Figure 4b). The genes that could not ne classified into any modules were collected in the gray module and were not used in following analysis. Then, the correlations between modules and clinical phenotypes (cancer and normal) were calculated and plotted as shown in Figure 5. The result shows that the blue module was highly related to cancer (r = 0.87, *p* = 4 × 10^−71^). Based on the cut-off criteria (|MM| > 0.8 and |GS| > 0.2), 147 module hub genes were selected in the blue module for further analysis.

### 3.3. Module Genes Identification and Functional Enrichment Analysis

In order to investigate the biological function of 147 candidate module genes, we performed GO and KEGG pathway enrichment analysis by using R package *clusterProfiler* (Figure 6a,b). Specifically, the BP group genes were focused on nuclear division, organelle fission, cell cycle checkpoint, and nuclear chromosome segregation. In addition, the CC group genes were mainly related to midbody, spindle, chromosomal region, and microtubule. Moreover, the MF group genes were mainly enriched in protein serine/threonine kinase activity, histone kinase activity, and cyclin-dependent protein kinase activity. Under the threshold of *p* < 1 × 10^−4^, the KEGG pathway analysis of 147 candidate module genes showed that cell cycle, progesterone-mediated oocyte maturation, and DNA replication were most significantly enriched.

### 3.4. PPI Network Construction and Hub Genes Identification

Among 147 candidate module genes, the PPI network was constructed with 145 nodes and 2205 edges using the STRING database, and visualized using Cytoscape software (Figure 7). Using the MCC algorithm of CtyoHubba plugin in Cytoscape, the co-expression network of the top ten highest-scored genes, including PDZ Binding Kinase (PBK), DNA Topoisomerase II Alpha (TOP2A), Cell Division Cycle Associated 8 (CDCA8), Abnormal Spindle Microtubule Assembly (ASPM), Cyclin A2 (CCNA2), Kinesin Family Member 20A (KIF20A), BUB1 Mitotic Checkpoint Serine/Threonine Kinase (BUB1), Aurora Kinase B (AURKB), Cyclin Dependent Kinase 1 (CDK1), and Cyclin B2 (CCNB2), is shown in Figure 8 and selected as hub genes.

### 3.5. Select Significant TNBC Genes Using LASSO Regression Model

This study adjusts for starting a variable selection with a large number of features by validation followed by training. Therefore, 147 candidate model genes were used to fit the LASSO regression model, and 10-fold cross-validation was used to train the model. According to lambda 1se, 0.03422616 was determined as an appropriate λ value (Figure 9A). Finally, six non-zero coefficient TNBC genes (CDK1, CENPF, MCM7, PACC1, TUBB, and UBE2C) were obtained (Figure 9B), which were then used in survival analysis to verify the prognostic value.

### 3.6. Survival Analysis and Prognostic Value Verification of Key Genes

In order to verify the prognostic value of 10 hub genes and six significant genes, KM Plotter was applied to 201 TNBC patients for OS analysis, and 576 TNBC patients for DFS analysis. Among the 10 pivot genes, the survival rate of BUB1 (FDR = 0.041) in the OS analysis was significantly different between the high and low expression groups (*p* < 0.05, Figure 10c). In the DFS analysis of BUB1 (FDR = 0.043), CCNA2 (FDR = 0.043), and CDCA8 (FDR = 0.043), there was a significant difference in survival rate between the high and low expression groups (*p* < 0.05, Figure 11). On the other hand, in the OS analysis of six important genes, the low expression of CENPF (FDR = 0.041) was significantly correlated with the poor survival rate of TNBC patients (*p* < 0.05, Figure 12). The DFS analysis showed that the high expression of PACC1 (FDR = 0.0152) was significantly correlated with the poor prognosis of TNBC patients (*p* < 0.05, Figure 13).

## 4. Discussion

In this study, we used a comprehensive bioinformatics analysis to identify 147 co-expressed genes that were differentially expressed between TNBC and normal tissue samples in the TCGA-BRCA dataset. As shown by the GO analysis results, 147 differentially co-expressed genes are mainly enriched in nuclear division, chromosomal region, and ATPase activity, while the KEGG pathway analysis is mainly enriched in the cell cycle signaling. Moreover, according to the MCC scores using CytoHubba plugin in Cytoscape, the top ten hub genes (PBK, TOP2A, CDCA8, ASPM, CCNA2, KIF20A, BUB1, AURKB, CDK1, and CCNB2) related to TNBC were screened out, and using LASSO regression feature selection screened six significant TNBC genes (CDK1, CENPF, MCM7, PACC1, TUBB, and UBE2C). We further evaluated the prognostic values of the ten highest-scored genes and six significant genes in KM Plotter databases. The OS analyses show that BUB1 and CENPF had a significantly poor prognosis in TNBC patients, while BUB1, CCNA2, CDCA8, and PACC1 show a significantly poor prognosis in TNBC patients in DFS analyses.

CENPF is reported to be a protein that interacts with microtubules and participates in the development of the cell cycle [23,24]. High expression of CENPF has been observed in various cancers, such as prostate cancer and breast cancer. In addition, the high expression of CENPF may eventually induce bone metastasis of breast cancer cells [25]. Several evidences suggest that enhanced CENPF is a reliable prognostic indicator of poor survival for breast cancer [25], prostate cancer [26], and hepatocellular carcinoma [27]. However, our research results show that lower expression of CENPF can cause poor OS prognosis in TNBC. It would be necessary to perform experiments to explore the mechanisms where CENPF is involved in patients with poor survival outcomes.

BUB1, a mitotic checkpoint serine/threonine kinase, serves an important role in the establishment of mitotic spindle checkpoint in breast cancer cells [28,29,30]. BUB1 expression is correlated with unfavorable prognosis in patients with malignant tumors, including breast [31,32] and liver cancers [33]. Our study suggested that a low level of BUB1 expression had reduced OS rates in TNBC patients. Nevertheless, in the DFS prognosis analysis, higher BUB1 expression had a poor survival rate compared with it expressing a low level. This would be worth investigating further.

Cell cycle regulator cyclin A2, also known as CCNA2, is a member of the highly conserved cyclin family. It has been confirmed to be up-regulated in a variety of cancers and participate in the progression of various cancers [34,35,36]. According to previous reports that the high expression of CCNA2 can lead to the production of TNBC cells, it has been identified as a potential candidate for the treatment of TNBC [37,38]. As we found in our study, the high expression of CCNA2 will promote better survival rates of DFS in patients with TNBC, which is worthy of in-depth study.

CDCA8 is known as Borealin/Dasra B, which is a component of the chromosomal passenger complex essential for transmission of the genome in mitosis [39]. CDCA8 plays an important role in several types of cancer, and its overexpression may act as an oncogene [40,41,42]. Overexpression of CDCA8 in breast cancer was reported to be associated with triple-negative phenotype and a shorter overall survival [43]. However, our study has shown that low CDCA8 expression results in shorter DFS survival. Further studied are required to investigate whether CDCA8 is an indicator of good prognosis.

The alias of PACC1 is called TMEM206, which is a member of the transmembrane (TMEM) protein family. The TMEM protein family is a potential player in the metastasis of cancer cells [44]. It is currently known that TMEM206 activates the AKT and ERK signaling pathways to worsen the progression of colorectal cancer and promote the proliferation, spread, and metastasis of colon cancer cells [45]. TMEM206 cells are also up-regulated in liver cancer cells [46], but the mechanism of action between TMEM206 and cancer is still poorly understood, and it has not been reported to be related to TNBC cells. The results of this study show that in the DFS analysis of PACC1 in TNBC patients, the high expression of PACC1 will lead to significant adverse prognostic effects, which may be related to the high expression of PACC1 inducing the proliferation and metastasis of cancer cells. Therefore, PACC1 may be a novel potential therapeutic target for TNBC.

This study still has its limitations, the unavoidable issue of batch effects, and the results may not apply to all TNBC patients. In summary, this study found BUB1, CCNA2, and PACC1 genes as three poor prognostic genes related to TNBC cells. These key genes may serve as potential biomarkers for the treatment or early diagnosis of TNBC.

## 5. Conclusions

The combination of WGCNA and differential gene expression analysis was used to construct a gene co-expression network, and the modular hub genes essential to TNBC were determined. Our results show that these genes are related to angiogenesis, cell proliferation, and cell cycle progression. In addition, we also used survival analysis to verify the results of MCC score ranking and LASSO feature selection, and found that the most important central gene in OS is BUB1, and the most important central genes in DFS are BUB1, CCNA2, and PACC1. However, the molecular mechanisms of these three key genes in TNBC need to be further studied through experiments, which will bring new insights to the research of TNBC.

## Figures and Tables

**Figure 1 genes-13-00902-f001:**
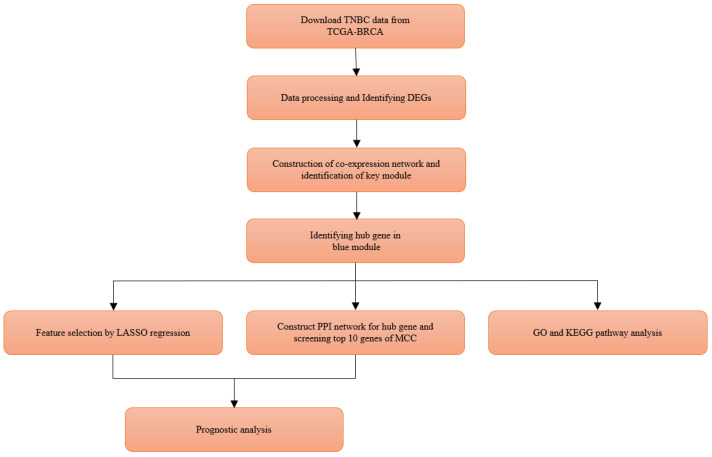
Flowchart of this study.

**Figure 2 genes-13-00902-f002:**
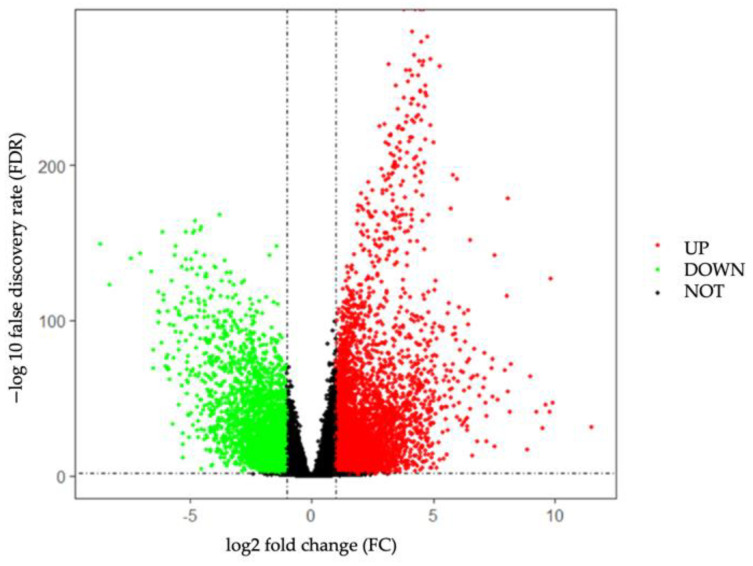
Volcano plot of DEGs between 116 TNBC and 113 normal tissue samples.

**Figure 3 genes-13-00902-f003:**
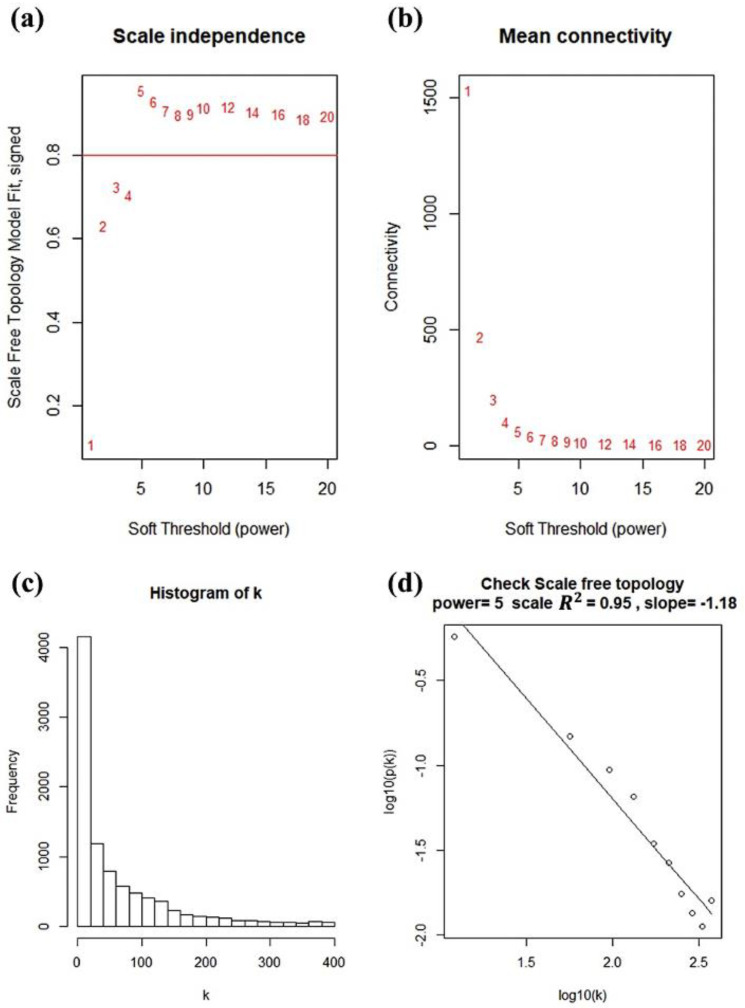
Determination of soft thresholding power in WGCNA. (**a**) Analysis of the scale-free fit index for various soft thresholding powers β. (**b**) Analysis of the mean connectivity for various soft thresholding powers. (**c**) Histogram of connectivity distribution when β = 5. (**d**) Checking the scale free topology when β = 5.

**Figure 4 genes-13-00902-f004:**
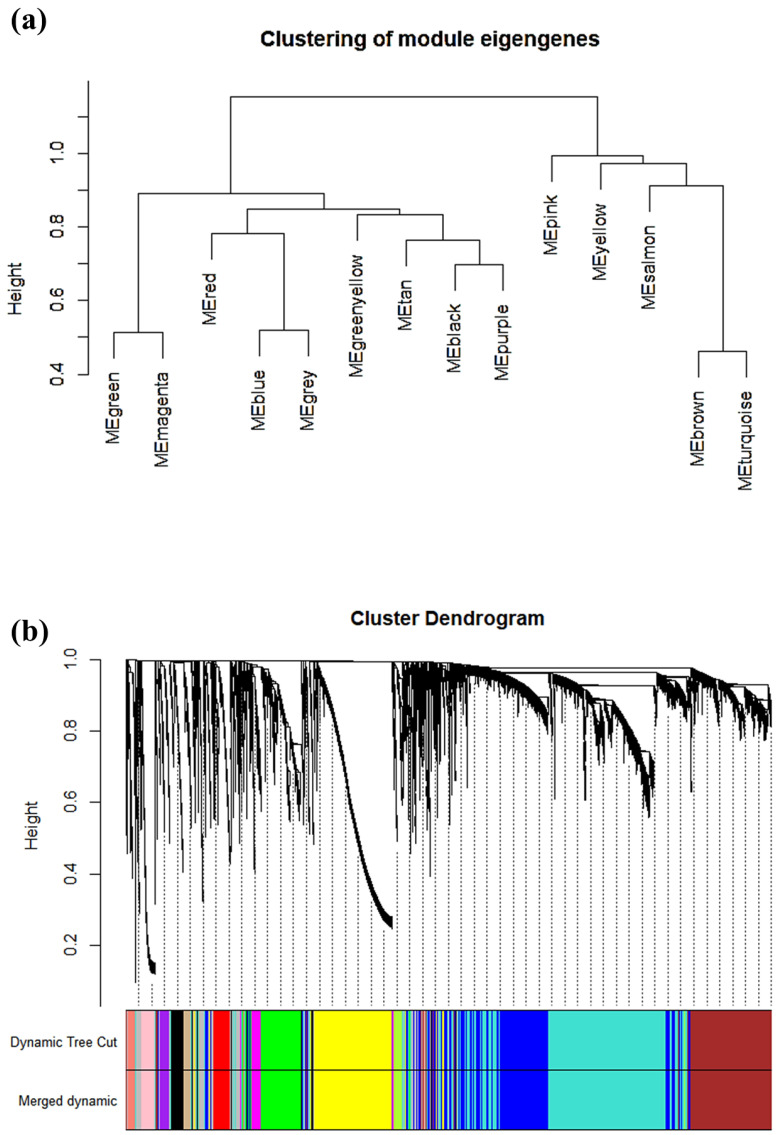
Module clustering results by WGCNA. (**a**) Dendrogram of dissimilarity between modules. (**b**) Gene dendrogram based on dissimilar (1-TOM) clustering. Each branch of the tree diagram represents a gene.

**Figure 5 genes-13-00902-f005:**
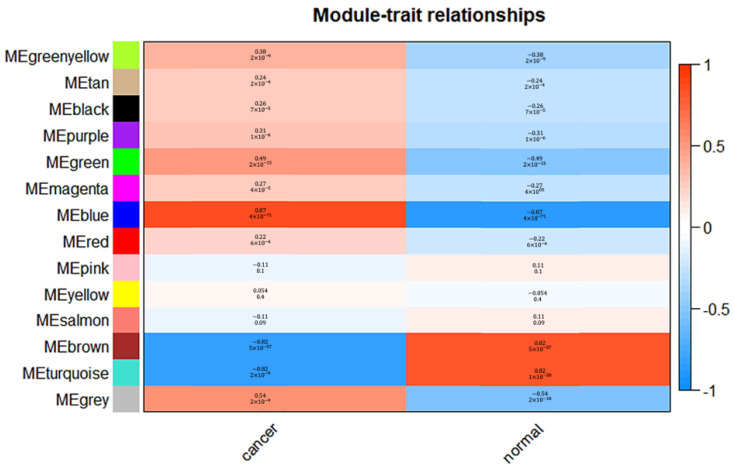
Correlation between modules and clinical features. (Each line contains the corresponding correlation and *p*-value.).

**Figure 6 genes-13-00902-f006:**
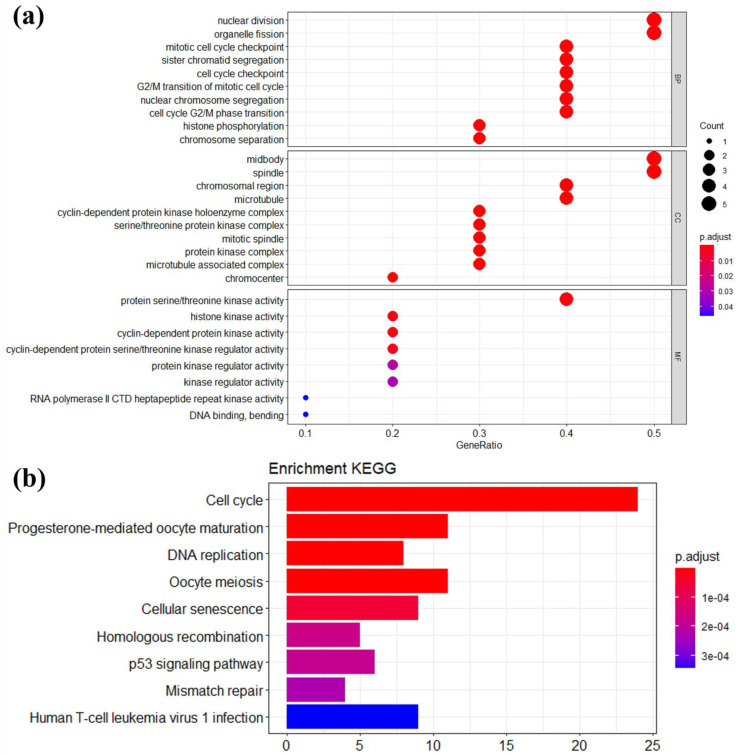
Functional enrichment analysis of 147 module genes. (**a**) GO analysis for hub genes. (**b**) KEGG pathway analysis for hub genes. BP, biological process; CC, cellular component; MF, molecular function.

**Figure 7 genes-13-00902-f007:**
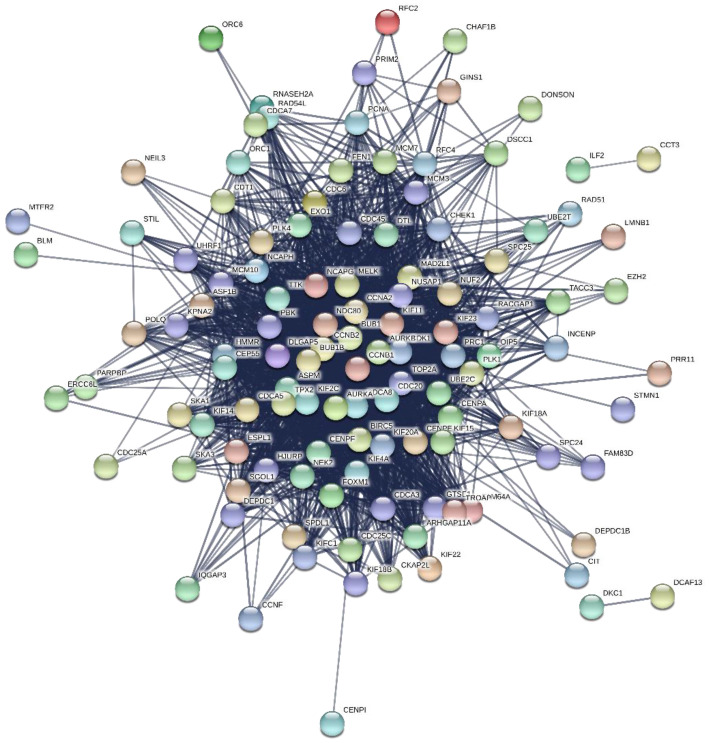
Visualization of the PPI network by Cytoscape.

**Figure 8 genes-13-00902-f008:**
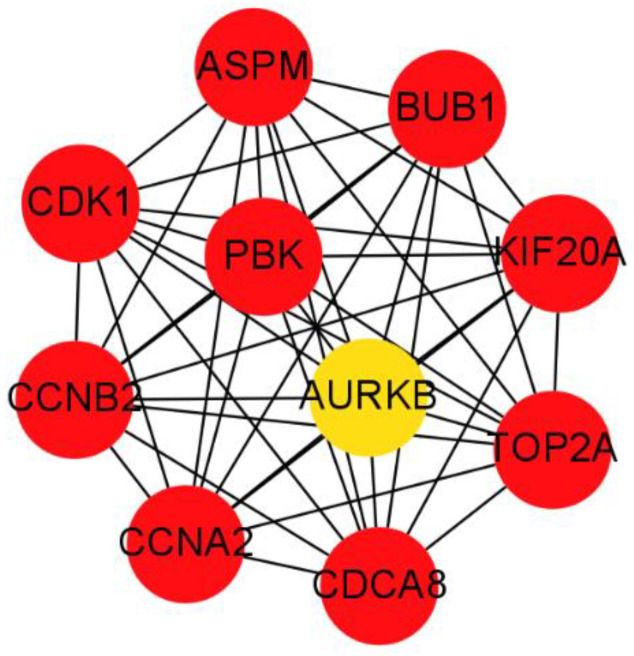
The top 10 genes with the highest value of MCC scores in the PPI network. (The closer the background color of the circle is to red, the higher the MCC score of the gene in the network, and the red is the highest score.).

**Figure 9 genes-13-00902-f009:**
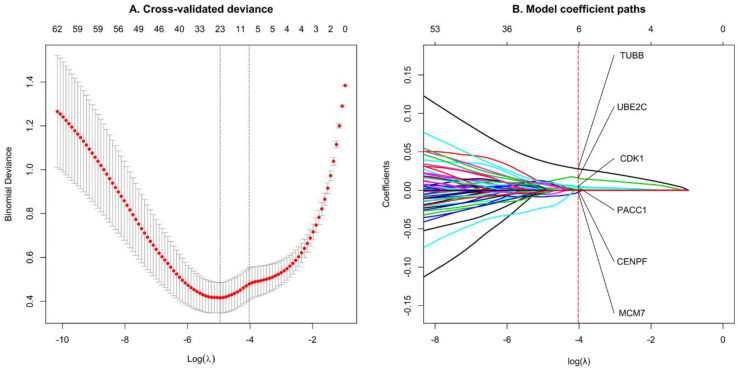
Feature selection was using the LASSO regression model through 10-fold cross-validation and lambda 1se. (**A**) Generate coefficient distribution plots for log(lambda) sequence. (**B**) LASSO non-zero coefficient 6 significant genes in TNBC.

**Figure 10 genes-13-00902-f010:**
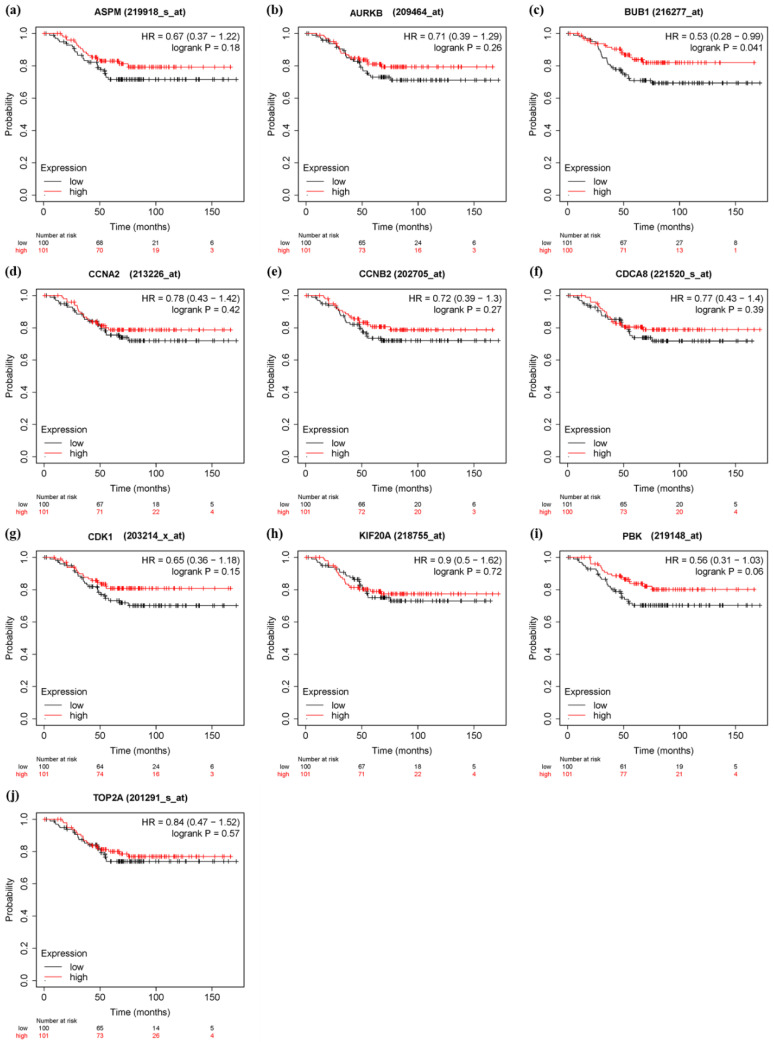
Overall survival (OS) analysis of 10 hub genes in TNBC patients by KM Plotter. (*n* = 201) (**a**) ASPM (**b**) AURKB (**c**) BUB1 (**d**) CCNA2 (**e**) CCNB2 (**f**) CDCA8 (**g**) CDK1 (**h**) KIF20A (**i**) PBK (**j**) TOP2A.

**Figure 11 genes-13-00902-f011:**
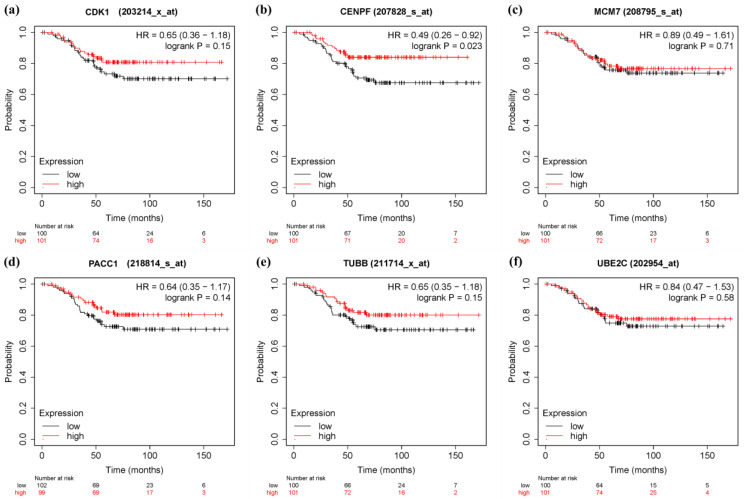
Overall survival (OS) analysis of 6 Significant genes in TNBC patients by KM Plotter. (*n* = 201) (**a**) CDK1 (**b**) CENPF (**c**) MCM7 (**d**) PACC1 (**e**) TUBB (**f**) UBE2C.

**Figure 12 genes-13-00902-f012:**
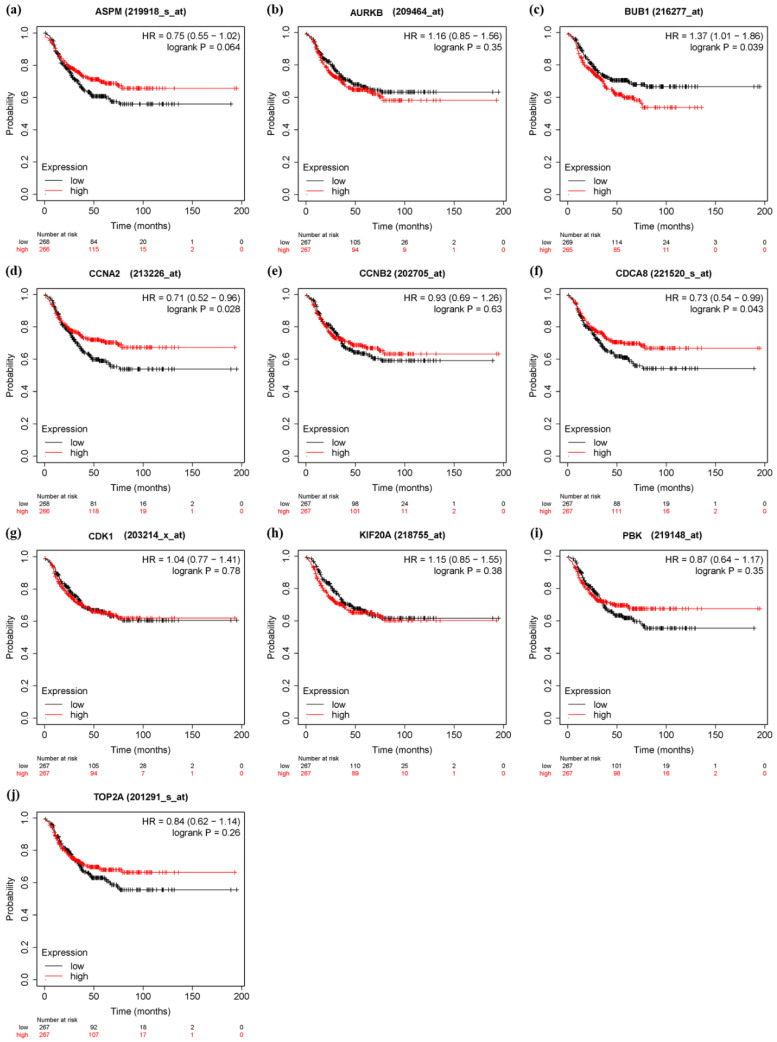
Disease free survival (DFS) analysis of 10 hub genes in TNBC patients by KM Plotter. (*n* = 534) (**a**) ASPM (**b**) AURKB (**c**) BUB1 (**d**) CCNA2 (**e**) CCNB2 (**f**) CDCA8 (**g**) CDK1 (**h**) KIF20A (**i**) PBK (**j**) TOP2A.

**Figure 13 genes-13-00902-f013:**
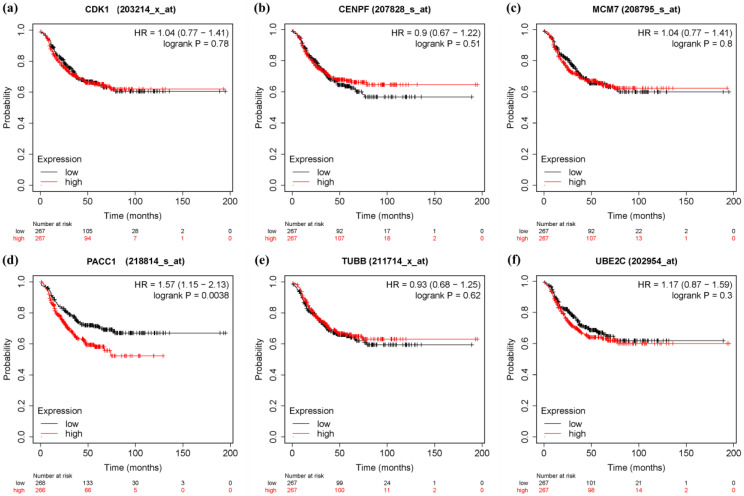
Disease free survival (DFS) analysis of 6 Significant genes in TNBC patients by KM Plotter. (*n* = 534) (**a**) CDK1 (**b**) CENPF (**c**) MCM7 (**d**) PACC1 (**e**) TUBB (**f**) UBE2C.

## Data Availability

The primary and processed data used in analysis can be downloaded from TCGA-BRCA publication page (https://portal.gdc.cancer.gov/) (accessed on 18 December 2021).

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
