# Peer review of "Identification of Key Prognostic Genes of Triple Negative Breast Cancer by LASSO-Based Machine Learning and Bioinformatics Analysis"

_genes, 2022, doi:10.3390/genes13050902_

Round 1
Reviewer 1 Report
This article is interesting and has the potential for a better understanding of the molecular mechanisms of triple negative breast tumors. However, I found some conflicting data that need to be further clarified. Below are some doubts. The authors cite some genes that may have a biological significance in this disease. See the discussion that begins on line 273. The CENPF gene is highly expressed in various cancers and is even related to bone metastases. But the results of this article showed that low levels of expression causing even worse prognosis. Is this not a disagreement with what is described in the literature? Could more articles be cited that show that what this conclusion is correct? Another similar case in the same paragraph includes the Bub1e CCNA2 gene in which, in this case, classics oncogenes that in the evaluated samples acted in a way to provide a better survival rate. For me this is quite conflicting and needs to be further discussed. Otherwise, I keep thinking that the samples evaluated did not have the necessary representative means in terms of gene expression and the final conclusion is wrong. The same occurs with the CDCA8. I am not confident that these genes are indeed reliable markers, so this needs to be better identified.
Author Response
Dear Reviewer,
Thank you for giving us the opportunity to improve and resubmit our manuscript titled “Identification of Key Prognostic Genes of Triple Negative Breast Cancer by LASSO-Based Machine Learning and Bioinformatics Analysis.” We have carefully studied the comments and suggestions, and then revised the manuscript accordingly. The changed and added texts in the manuscript are shown in red. Also, please note that all results and figures have been changed slightly, which are different from that in the original one, because we added the one new dataset and the analysis methods. We hope that the revision could be acceptable, and that our responses adequately address the comments. Should you have any questions, please contact us without hesitation.
Below is a point-by-point response to the reviewer’ comments.
Comments 1:
The authors cite some genes that may have a biological significance in this disease. See the discussion that begins on line 273. The CENPF gene is highly expressed in various cancers and is even related to bone metastases. But the results of this article showed that low levels of expression causing even worse prognosis. Is this not a disagreement with what is described in the literature? Could more articles be cited that show that what this conclusion is correct? Another similar case in the same paragraph includes the Bub1e CCNA2 gene in which, in this case, classics oncogenes that in the evaluated samples acted in a way to provide a better survival rate. For me this is quite conflicting and needs to be further discussed. Otherwise, I keep thinking that the samples evaluated did not have the necessary representative means in terms of gene expression and the final conclusion is wrong. The same occurs with the CDCA8. I am not confident that these genes are indeed reliable markers, so this needs to be better identified.
Response:
We would like to thank the reviewer for the thoughtful comments. To be less assertive, the sentences which were pointed out by reviewer have been modified as follows in our revised manuscript:
- “It would be necessary to perform experiments to explore the mechanisms that CENPF is involved in patients with poor survival outcomes.” (pages 14, lines 279-280)
- “Further studied are required to investigate whether CDCA8 is an indicator of good prognosis.” (pages 14, lines 300-301)
To the best of our knowledge, BUB1, CCNA2 and PACC1 expression have been proven to be correlated with unfavorable prognosis in patients with malignant tumors such as breast cancer, which is consistent with our finding. Thus, we have made revisions in both discussion and conclusion accordingly (pages 1, lines 25; pages 15, lines 323-324). We hope the manuscript has been improved towards your concerns after this revision.

Reviewer 2 Report
In this manuscript, the authors have performed the comprehensive bioinformatic analysis among TNBC and normal samples to identify 10 hub genes and 6 significant genes. Furthermore, they evaluated the prognostic role of these genes. However, there are some remaining questions to be answered:
1, For overall survival (OS) and disease free survival (DFS) analysis, the pattern of BUB1 (Fig 10c and Fig 11c) and PACC1 (Fig 12d and Fig 13d) were opposite between OS and DFS. Could the authors discuss why different expression of BUB1 and PACC1 exhibited opposite pattern between OS and DFS.
2, For selected genes, only BUB1 showed prognosis role in both OS and DFS. Some genes show prognosis role in OS, not DFS, vise versa. Could the authors discuss the possible reasons?
3, The authors need to check the text editing. For example, Line 260, extra “-” in “differentially”. Line 265, extra space after “out, and”. Line 268, extra space after “genes in”. Missing “T” after “Plotter database.”
Author Response
Dear Reviewer,
Thank you for giving us the opportunity to improve and resubmit our manuscript titled “Identification of Key Prognostic Genes of Triple Negative Breast Cancer by LASSO-Based Machine Learning and Bioinformatics Analysis.” We have carefully studied the comments and suggestions, and then revised the manuscript accordingly. The changed and added texts in the manuscript are shown in red. Also, please note that all results and figures have been changed slightly, which are different from that in the original one, because we added the one new dataset and the analysis methods. We hope that the revision could be acceptable, and that our responses adequately address the comments. Should you have any questions, please contact us without hesitation.
Below is a point-by-point response to the reviewer’ comments.
Comments 1:
For overall survival (OS) and disease free survival (DFS) analysis, the pattern of BUB1 (Fig 10c and Fig 11c) and PACC1 (Fig 12d and Fig 13d) were opposite between OS and DFS. Could the authors discuss why different expression of BUB1 and PACC1 exhibited opposite pattern between OS and DFS.
Response:
Considering the meaning of OS and DFS from the perspectives of clinical traits and clinical practice, we might think that different patterns between OS and DFS are due to the trial design for time-to-events endpoints. OS is defined as the time from randomization to death from any cause, while DFS refers to the time from randomization to the first event of either recurrent disease or death. However, this assumption still needs to be confirmed by large-scale randomized controlled studies.
Comments 2:
For selected genes, only BUB1 showed prognosis role in both OS and DFS. Some genes show prognosis role in OS, not DFS, vise versa. Could the authors discuss the possible reasons?
Response:
Survival rates depend on many different factors. As mentioned earlier, one of the possible reasons that could affect the survival is the difference in trial design between OS and DFS. In our opinion, all of patients are not exactly alike and response to treatment also varies from one to another. Also, another factor we think may be the grade and/or stage of cancer cell. It could affect prognosis of patients with breast cancer.
Comments 3:
The authors need to check the text editing. For example, Line 260, extra “-” in “differentially”. Line 265, extra space after “out, and”. Line 268, extra space after “genes in”. Missing “T” after “Plotter database.”
Response:
Thanks for your kind reminders. We have modified typos in the manuscript accordingly. (pages 14, lines 260, 265, 268)
